# Production of [^11^C]Carbon Labelled Flumazenil and *L*-Deprenyl Using the iMiDEV™ Automated Microfluidic Radiosynthesizer

**DOI:** 10.3390/molecules27248843

**Published:** 2022-12-13

**Authors:** Hemantha Mallapura, Laurent Tanguy, Bengt Långström, Ludovic Le Meunier, Christer Halldin, Sangram Nag

**Affiliations:** 1Department of Clinical Neuroscience, Center for Psychiatry Research, Karolinska Institutet and Stockholm County Council, 17176 Stockholm, Sweden; 2PMB Alcen, Route des Michels CD56, F-13790 Peynier, France; 3Department of Chemistry, Uppsala University, 75236 Uppsala, Sweden

**Keywords:** PET radiotracers, microfluidics, iMiDEV™, [^11^C]flumazenil, [^11^C]*L*-deprenyl, radiosynthesis, microfluidic cassette, dose-on-demand (DOD)

## Abstract

In the last decade, microfluidic techniques have been explored in radiochemistry, and some of them have been implemented in preclinical production. However, these are not suitable and reliable for preparing different types of radiotracers or dose-on-demand production. A fully automated iMiDEV™ microfluidic radiosynthesizer has been introduced and this study is aimed at using of the iMiDEV™ radiosynthesizer with a microfluidic cassette to produce [^11^C]flumazenil and [^11^C]*L*-deprenyl. These two are known PET radioligands for benzodiazepine receptors and monoamine oxidase-B (MAO-B), respectively. Methods were successfully developed to produce [^11^C]flumazenil and [^11^C]*L*-deprenyl using [^11^C]methyl iodide and [^11^C]methyl triflate, respectively. The final products 1644 ± 504 MBq (*n* = 7) and 533 ± 20 MBq (*n* = 3) of [^11^C]flumazenil and [^11^C]*L*-deprenyl were produced with radiochemical purities were over 98% and the molar activity for [^11^C]flumazenil and [^11^C]*L*-deprenyl was 1912 ± 552 GBq/µmol, and 1463 ± 439 GBq/µmol, respectively, at the end of synthesis. All the QC tests complied with the European Pharmacopeia. Different parameters, such as solvents, bases, methylating agents, precursor concentration, and different batches of cassettes, were explored to increase the radiochemical yield. Synthesis methods were developed using 3–5 times less precursor than conventional methods. The fully automated iMiDEV™ microfluidic radiosynthesizer was successfully applied to prepare [^11^C]flumazenil and [^11^C]*L*-deprenyl.

## 1. Introduction

Positron emission tomography (PET) is an advanced, indispensable, sensitive, and non-invasive nuclear medicine medical imaging technique used in various medical applications. It allows for visualizing and quantifying brain pathology (fibrillar Aβ, tau, activated microglia and astrocytosis) and functional changes (cerebral glucose metabolism, neurotransmitter and neuroreceptor activity). Therefore, PET has become a valuable molecular imaging modality with several applications in oncology [1], neuroscience [2,3,4] and cardiovascular imaging [5]. The transfer of tracer methods from the basic biological sciences to humans with PET is made possible by the unique nature of radionuclides such as fluorine-18 (^18^F), carbon-11 (^11^C), zirconium-89 (^89^Zr), copper-64 (^64^Cu), and gallium-68 (^68^Ga). In the recent decade, PET imaging has become an essential tool in drug discovery [6,7,8].

[^18^F]fluorodeoxyglucose ([^18^F]-FDG) is one of the most produced and used PET tracers in molecular imaging applications. This is due to in vivo biochemistry use, the long half-life of the ^18^F isotope (110 min), the high positron decay ratio (97%), and its positron energy (maximum 0.635 MeV). The [^18^F]FDG is produced on a large scale with conventional radiosynthesizers and then distributed into single/multiple doses to the nearest PET centers or used in-house. However, this model is not viable for producing radiotracers with short-lived isotopes such as carbon-11 (20.4 min) and gallium-68 (67.8 min) that are utilized for clinical applications. The production expenditure from this infrastructure is high to produce a single dose or dose-on-demand (DOD) [9]. Other limitations of conventional synthesizers are consuming of more reagents and precursors, longer synthesis time, lower molar activity and increased risk of higher radioactive exposure to personnel.

To overcome the limitations of conventional radiosynthesizers and to reduce the production cost, we needed a de-centralized single dose or DOD model. In this direction, lab on a chip or microfluidic technique models look promising [10,11,12]. There are many microfluidic techniques have been explored for the development of efficient radiolabeling methods, mainly divided into 3 categories (i) continuous flow or channel flow, (ii) batch type or chamber, and (iii) digital microfluidics or microdroplets. These microfluidic techniques have advantages and limitations as well [13]. The amount of chemicals can be reduced 10 to 100-fold using a digital or droplet microfluidic radiochemical technique. Microreactor and microfluidic channels are advantageous for exploring and performing continuous flow of reaction mixtures to perform radiochemical reactions [14]. Some microfluidic synthesizers have been utilized for in-house clinical and preclinical applications [10,15,16]. However, the reliability and reproducibility of radiotracers from these microfluidic synthesizers are still challenging. Thus, there is a need to explore reliable and reproducible microfluidic techniques to synthesize PET radioligands using microfluidic technology. Compared with continuous flow and microdroplet microfluidics, the batch-type microfluidics technique looks promising [13]. It allows a controlled volume to perform multiple step reactions and is easily adaptable to a single dose/DOD model. In summary, it is a miniature version of a conventional radiosynthesizer with microfluidic benefits.

We have introduced a batch-type fully automated microfluidic iMiDEV™ synthesizer and synthesized sodium [^18^F]fluoride ([^18^F]NaF) as an initial proof of concept study [15]. This radiochemistry module is microfluidic cassette-based and compatible with liquid and gas-phase reactions to produce diversified radiotracers for preclinical and clinical applications. In radiopharmaceutical drug synthesis, several microfluidic techniques have been explored with ^18^F and ^68^Ga radionuclides but not with ^11^C radionuclides [17,18,19,20,21,22,23]. There are many applications of ^11^C radioligands in neuroimaging and drug discovery [8,24,25,26]. We selected two classical ^11^C radioligands, [^11^C]flumazenil [27,28] and [^11^C]*L*-deprenyl [29,30,31], to explore radiochemical conversion by using different solvents and bases at room temperature reactions.

This study aimed to (i) synthesize, purify, and formulate ^11^C radiotracers such as [^11^C]flumazenil and [^11^C]*L*-deprenyl with the iMiDEV™ automated synthesizer using a microfluidic cassette and (ii) adapt selected tracer production in a single dose or DOD model.

## 2. Results

### Automated Radiosynthesis of [^11^C]Flumazenil and [^11^C]L-Deprenyl on an iMiDEV™ Microfluidic Cassette

In-target produced [^11^C]methane ([^11^C]CH_4_) was used to synthesize [^11^C]methyl iodide ([^11^C]CH_3_I) or [^11^C]methyl triflate ([^11^C]CH_3_OTf) for all performed syntheses. [^11^C]CH_3_I was utilized for the synthesis of [^11^C]flumazenil, whereas [^11^C]CH_3_OTf was utilized for [^11^C]*L*-deprenyl. The radiochemical reaction was performed on R1, and solid-phase extraction (SPE) purification was performed on R4. For [^11^C]flumazenil synthesis, 200 µg precursor in 100 µL of a 1:1 mixture of dimethyl sulfoxide (DMSO) and dimethylformamide (DMF) with potassium hydroxide (KOH) powder (4–5 mg) was used, and the reaction time was 3 min. Pentamethyl piperidine (PMP) mixture (2:1 mixture of methanol (CH_3_OH) and acetonitrile (CH_3_CN) with PMP base) was used for [^11^C]*L*-deprenyl, and the reaction was spontaneous with [^11^C]CH_3_OTf. The yield of radiochemical conversion for [^11^C]flumazenil and [^11^C]deprenyl was 22–28%. The detailed production of radioligands in this study is summarized in Table 1. [^11^C]flumazenil was produced at an activity of 1644 ± 504 MBq (*n* = 7) with radiochemical purity >99.00%, and [^11^C]deprenyl was produced at an activity of 533 ± 10 MBq (*n* = 3) with radiochemical purity >97.7% (Appendix A). The total volume of the final product was around 8 mL. The molar activities for [^11^C]flumazenil and [^11^C]*L*-deprenyl were 1912 ± 552 GBq/µmol and 1463 ± 439 GBq/µmol at the end of synthesis (EOS), respectively. The total synthesis time, including high performance liquid chromatography (HPLC) purification and formulation, was 34 min for [^11^C]flumazenil and 39 min for [^11^C]deprenyl after delivery of [^11^C]CH_4_ from the cyclotron. All the QC tests were complied with the European Pharmacopeia and are summarized in Table 2.

## 3. Discussion

In the iMiDEV™ radiosynthesizer, the microfluidic cassette replaces a conventional glass reactor with a built-in microreactor. The tubing was replaced with microfluidic channels, and macro SPE cartridges were replaced with micro cartridges. In our previous publication [15], the liquid phase for the reaction on R1 and R3 was explored by synthesizing [^18^F]NaF. However, in this study, we explored a gas phase reaction at room temperature for the first time on a microfluidic cassette, and reactions were performed on a micro cartridge filled with C18 beads with a capacity of 50 µL and integration of preparative HPLC into the module.

The radiosynthesis of [^11^C]flumazenil and [^11^C]*L*-deprenyl was performed using the automated iMiDEV microfluidic synthesizer with a microfluidic cassette. The produced yields (1644 ± 504 MBq of [^11^C]flumazenil and 533 ± 10 MBq of [^11^C]*L*-deprenyl) are compatible with single-dose or DOD production, which is sufficient to be utilized for preclinical applications. Reactor R1 and R3 were examined for initial test runs, but only R1 allows for the reuse of cassettes, unlike R3, which wets the vent after the first use. For gas-phase reactions, microfluidic cassettes should be free from water tracers before performing the reaction on selected microfluidic channels that are used for radioactive gas transfer ([^11^C]CH_3_I and [^11^C]CH_3_OTf). Considering the accessibility of microfluidic channels and the reuse of cassettes, only R1 was used to perform radiosynthesis. As a result, R1 was optimized for radioactivity trapping and radiochemical conversion. The manual operational mode was used during the optimization of the synthesis. After the reaction conditions were optimized, the entire synthesis was performed in a fully automated mode.

Different solvents were used to influence radiochemical conversion (RCC) for [^11^C]flumazenil (Appendix A) and [^11^C]*L*-deprenyl (Appendix A). The synthesis methods for [^11^C]flumazenil and [^11^C]*L*-deprenyl were already established with conventional radiosynthesizers. Polar aprotic solvents (acetonitrile, acetone, DMSO, and DMF) were used with [^11^C]CH_3_I or [^11^C]CH_3_OTf as a methylating agent. Therefore, we explored the same solvents and reagents for initial tests with microfluidic cassettes.

For [^11^C]flumazenil synthesis, when acetone was used as a reaction solvent, the trapping efficiency of [^11^C]CH_3_OTf on R1 was low and yielded low radiochemical conversion (1–5%). Different flow rates (5–15 mL/min) of the process gas (helium) were explored to improve the residence time of the methylating agent on R1. However, it did not improve the trapping efficiency of [^11^C]CH_3_OTf. We assumed that because of the lower volume of R1 (50 µL), acetone evaporated during [^11^C]CH_3_OTf transfer and left the precursor dry on R1 which we attributed this to reaction failure. The same experience was reported when acetone was used as a solvent in on-column continuous flow synthesis [32].

Different solvents and bases were used to optimize the trapping of methylating agents and radiochemical conversion. Reaction solvents with higher viscosities and boiling points than acetone were explored. DMF and DMSO solvents were used with sodium hydroxide (NaOH) as a base, and the methylation reaction with [^11^C]CH_3_OTf was poor. Thus, the methylating agent was changed to [^11^C]CH_3_I and used to perform tests with different solvents and NaOH as a base. By referring to the simple captive solvent method [33], DMF and DMSO were tested as solvents and performed reactions, but this did not work for the microfluidic cassette, possibly because of the lower concentration of precursor (2 µg/µL) compared to the captive solvent method (7.5 µg/µL).

Later, the base was changed from aqueous NaOH to KOH powder with DMSO. This resulted in 30% trapping of [^11^C]CH_3_I and an improvement in radiochemical conversion. The solvent was changed to a 1:1 mixture of DMSO and DMF with KOH powder to optimize the reaction conditions. This condition improved trapping, and radiochemical conversion was over 50%. After optimizing the base amount, precursor loading time on the reactor, and reaction time (1–5 min), we achieved over 90% radiochemical conversion. There was no significant difference in the radiochemical conversion with a reaction time of 3 to 5 min. Thus, the reaction time was set to 3 min.

For [^11^C]*L*-deprenyl, we adopted the well-established in-house radiosynthesis method to microfluidic cassettes with [^11^C]CH_3_OTf as a methylating agent and used the same solvents (PMP mixture; PMP base in acetonitrile and methanol) for the reaction. The radiolabeling conversion was 65–80% with a microfluidic cassette. However, after using different batches of cassettes, changes were observed in the radiolabeling conversion and release efficiency from R1 after HPLC injection. The reason was unknown, apart from different batches of cassettes. All the reagents, including the base, were changed to address this problem, yet the same problem persists. Thus, the synthesis yield of [^11^C]*L*-deprenyl was low (533 ± 10 MBq; *n* = 3).

The precursor was tested using a conventional glass vial as a reactor, and the radiochemical conversion was above 65%. [^11^C]CH_3_OTf was trapped on R1 by loading reaction solvents (MeOH and ACN) alone and with a PMP base. These tests also yielded the same results, and 25 to 40% of trapped radioactivity remained on R1. We decided to investigate trapping with different types of beads (glass and stainless steel beads) on R1. Due to the problem associated with the bead filling process, we did not perform any tests with these beads. The tests were conducted with [^11^C]CH_3_I with different solvents and bases to improve radiochemical yield (Appendix A). The trapping and radiochemical conversion were poor with these combinations.

### 3.1. Volume

For the reaction, 200–250 µL precursor solution was used from a 4 mL vial. These vials had more than the expected dead volume (80 to 100 µL). The R1 capacity is 50 µL and needs just above 50 µL to load the precursor into the R1. Because of the 4 mL vial, half of the precursor was left in the vial, and sometimes more. Additionally, the precursor was not loaded into reactor R1 due to the precursor mixture being concealed in the vial when it was inverted (above the neck) and vigorous cassette clamping. After experiencing these problems, the 4 mL vials were changed to 300 µL vials. With 300 µL vials, the precursor volume was decreased to 100 µL from 250 µL, and the dead volume was low. The pressure and time for precursor loading were adjusted with 300 µL vials. If the concentration of the precursor was kept unchanged, there was no influence of volume on radiochemical yield compared with 100 µL and 250 µL. However, not all the precursor volume (100 µL) was utilized for the radiochemical reaction, and yet this decreased the amount of precursor used in the synthesis by 2.5 times (200 µg for both [^11^C]flumazenil and [^11^C]*L*-deprenyl), and the reaction volume by 3–5 times. Further decreasing the precursor volume, we need to modify the existing vial or design a new one. This will be addressed in the near future.

### 3.2. The Concentration of the Precursor

The initial concentration of precursor was increased from 1 µg/µL to 3 µg/µL, but there was an improvement in radiochemical conversion with increased concentration. When the concentration was 2 µg/µL to 3 µg/µL, there were no significant improvements in the radiochemical conversion. The final concentration was optimized to 2 µg/µL. We analyzed the amount of precursor loaded on reactor R1 using the same concentration and volume used for the synthesis. Compared with the R1 size in terms of surface area and precursor volume, 30 to 40 µg of the precursor was utilized for the reaction, and the quantity of precursor was 8–10 times less than that which was used in conventional radiosynthesizers for regular clinical production [27,33,34,35]. The precursor quantity used for the synthesis was reduced almost 3–5 times compared to the conventional method. The entire synthesis was performed in automated mode using the iMiDEV module. We achieved relatively high molar activity of [^11^C]flumazenil and [^11^C]*L*-deprenyl compared to conventional radiosynthesizers.

### 3.3. Radiosynthesis with Microfluidic Cassette

The microfluidic cassette is an essential part of radiosynthesis. Apart from semi preparative HPLC purification, the remaining synthesis steps, such as radiochemical reaction, SPE purification, and formulation steps, are performed on the cassette. After optimization of all the synthesis conditions, complete radiosynthesis was performed on the microfluidic cassette. We noted that different batches of microfluidic cassettes varied in the efficiency of trapping radioactive gases such as [^11^C]CH_3_I and [^11^C]CH_3_OTf and radiochemical conversion (20 to 90%).

This result indicated variations in the cassettes concerning different batches, and our investigation found that variations in the density of beads on R1 and R4 significantly influenced the amount of final product and synthesis time. The arrangement of beads on the R1 chamber must be examined before it is used for radiosynthesis by performing the flow resistance measurement on R1. The higher flow resistance on R1 was detrimental to precursor and reagent transfer (flumazenil and deprenyl), thus the poor trapping of [^11^C]CH_3_I/[^11^C]CH_3_OTf and low radiochemical yield. Once we had optimized the pressure and time parameters across different batches of cassettes, we noted that issues with precursor loading and HPLC injection were due to flow resistance variations on R1. As a result, the precursor passed through the R1 too quickly or sometimes unevenly dispersed. Since the flow was linear, radioactive gas was erratically distributed when it was passing through R1. This phenomenon led to less surface area for radioactive gas (methylating agent) interaction with beads in R1, eventually the precursor. Consequently, it leads to lower trapping of [^11^C]CH_3_I or [^11^C]CH_3_OTf gas on R1 and thus, poor radiochemical conversion.

Our test results found optimized bead filling on R1 through the pressure drop test, and we kept the cassettes with similar pressure drops, which yielded good radiochemical conversion, and rejected the cassettes that did not meet the criteria. The experimental results were reproducible from cassettes with similar flow resistance on R1. In this direction, we need to investigate and improve the bead filling process to maintain a similar flow resistance on R1 and plan to investigate beads with a similar diameter to assess the radiochemical conversion.

The synthesis time varied from 29 to 40 min from cassette to cassette due to the flow resistance on R4. After collection by semi-preparative HPLC, the R4 was used for SPE purification of the product (flumazenil and deprenyl). For [^11^C]flumazenil, the synthesis time was 35 min, and [^11^C]*L*-deprenyl, the synthesis time was over 35 min, because R4 showed abnormally high flow resistance. This strongly limited the flowrate during the SPE purification and formulation steps of the process.

The microfluidic cassettes were reused for tests up to 10 times. Cassettes were cleaned with acetone prior to reuse. For research and development, the same cassette can be used multiple times. The radiochemical conversion of [^11^C]flumazenil and [^11^C]*L*-deprenyl from the used cassettes are similar to the new cassettes. Sometimes, we experienced clogging of microfluidic channels with salt from reagents such as buffer and saline when it was used more than a few times. To avoid clogging problems with microfluidic channels, those used with buffer or saline should be rinsed with water after the test synthesis.

The same cassette can be used for gas and liquid phase reactions, but suitable beads should be filled in reactors for reaction and SPE purification. This is an advantage of the microfluidic cassette, which can be utilized for various radiotracer syntheses with ^11^C, ^18^F, and ^68^Ga radionuclides. The cost of the microfluidic cassette and iMiDEV™ microfluidic radiosynthesizer are competitive with the existing cassette based radiosynthesizers on the market.

The current study was focused on exploring room temperature reaction with ^11^C gaseous radionuclide with the microfluidic cassette, and which is due to an associated problem with R2. Reactor 2 allows to perform high temperature reactions in the liquid phase, but not in the gas phase. Because the present reactor 2 design is not supportable to bubble or trap methylating agent, this is a limitation of microfluidic cassette to perform high temperature reactions in the gas phase. Despite this limitation, radiotracers were synthesized by exploring room temperature conditions to perform the radiochemical reaction.

## 4. Materials and Methods

All reagents and materials were used as they were received unless mentioned otherwise. Desmethyl-*L*-deprenyl (precursor) and *L*-deprenyl (reference standard) were synthesized in the in-house lab, and desmethyl flumazenil (precursor) and flumazenil (reference standard) were procured from Pharmasynth AS, Tartu, Estonia. PMB-Alcen (Peynier, France) furnishes microfluidic cassettes. C18 beads purchased from Waters, Rydalmere, Australia. Dimethyl sulfoxide (DMSO), dimethylformamide (DMF), potassium hydroxide (KOH), sodium hydroxide (NaOH), ammonium hydroxide (NH_4_OH), and methanol (CH_3_OH) were purchased from Sigma Aldrich, Germany. Pentamethyl piperidine (PMP) 95% was purchased from Thermo Scientific, Göteborg, Sweden. Sterile water and saline (0,9%) from Braun, Melsungen, Germany. Ethanol (99.5%) from KiiltoClean, Malmö, Sweden, and acetonitrile (CH_3_CN) (>99.8%) was purchased from Fisher Scientific, China. Sterile vent filters (Millex FG. 0.2 µm, 25 mm) and sterile filters (Millex GV, 0.22 µm, 33 mm) from Merck Millipore, Carrigtwohill, Ireland. 10 mL sterile product vials and phosphate-buffered saline from APL, Kungens Kurva, Sweden. Milli-Q water (18Mohm) was obtained from an in-house Millipore water purification system. SCHOTT ISO Clear Type/Tubular Glass vials of capacity 4 mL and 15 mL from the Nordic pack, Nykvarn, Sweden. Inserter vial (300 µL) from Thermo Scientific, Langerwehe, Germany. [^11^C]Methane ([^11^C]CH_4_) was produced on a PET Trace 16.4 MeV Cyclotron from General Electric (GE), Uppsala, Sweden.

### 4.1. The iMiDEV™ Automated Microfluidic Radiosynthesizer

iMiDEV™ (PMB-Alcen, Peynier, France) is a newly introduced batch-type reactor microfluidic automated radiosynthesizer. The working conditions of the radiosynthesizer and microfluidic cassette were elaborately presented previously [15], and the entire synthesis was performed on a microfluidic cassette. iMiDEV™ is the first sophisticated batch-type microfluidic automated radiosynthesizer with HPLC purification and formulation of the final product. iMiDEV™ will be integrated with the automated iMiLAB™ PET tracer production system after developing reliable synthesis methods. iMiLAB™ is a fully automated GMP robotic system that assists in performing PET tracer production by limiting in-person involvement in production.

The synthesizer is assembled with two different levels (Figure 1). The synthesis box and docking station (pneumatic valves system) are inside the hot cell. The computer with dedicated software and the semi-preparative HPLC system are located outside the hot cell. The synthesis box where the microfluidic cassette is installed and is interfaced with 34 microfluidic and 9 electro valves (EVs). These microfluidic valves are operated by compressed air, and the electro valves are used for transferring reagents from vials. Helium/nitrogen gas was used to transfer reagents to a different part of the cassette with a range of 0.1 to 1.7 bar.

The semi-preparative HPLC system is connected to the microfluidic cassette to allow the injection of the radiolabeled reaction mixture onto the HPLC column for purification. The semi-preparative column, UV cell, and radio detector are integrated into the synthesis box. The UV detector, mobile phases, and pump (up to 400 bar) will be kept outside and connected to the synthesis box. In the HPLC system, there are two Rheodyne valves and loops, and these are dedicated to automated injection and product collection.

All synthesis steps were performed by iMiDEV™ supervision (human–machine interface-HMI) software. The HMI (Figure 2) allows synthesis in different modes, such as manual, semi-automated, and fully automated. The automated mode performs all synthesis steps, including radiochemistry, HPLC purification, and the final product formulation.

Manual or semi-automated modes are used for the development and optimization process. There are separate tabs for pressure reading and the recipe loading. The supervision collects all the data during the synthesis, including valve operation, track pressure, radioactive sensors, and HPLC purification. After the synthesis, iMiDEV™ supervision generates a report (Appendix A) with all data and an Excel^®^ spreadsheet including the operator’s comments at the end of the synthesis. This report is essential for the root cause analysis of any issues arising during synthesis.

### 4.2. iMiDEV™ Microfluidic Cassette

The chemistry process, SPE purification, and formulation steps are performed on the microfluidic cassette (Figure 3), which is made of a cyclic olefin polymer (COP 1430R) suitable for medical and microfluidic applications and manufactured by miniFAB, Australia. It is channeled with microfluidic paths which are connected to 34 microfluidic valves and nine reagent vials (A to I from 100 µL to 15 mL). The valve positions are located under the cassette and sealed with a 150 µm thin cyclic olefin copolymer (COC-E-140) film, which allows for sealing the valve position with high pressure activated solenoid valves (Microfluidic valves 1 to 34).

The cassette has four reactors, and three of them (R1, R2, and R3) are used for chemistry. Reactor R1 and R3 with 50 µL capacity are used for room temperature reactions and isotope concentration supported by resin or polymer material. Reactor R2 with a capacity of 286 µL facilitates high-temperature reactions up to 130 °C. Reactor R4 with a 200 µL capacity is utilized for solid-phase extraction (SPE) purification. There are two embedded vents made with polytetrafluoroethylene (PTFE), which are placed before and after R2 to facilitate the transfer of radioactivity and precursor without creating bubbles in R2. The four mixers help to mix reagents when passing through them, and the liquid moves through microfluidic channels and mixers in laminar flow. The formulation chamber (12 mL) was used for the final product formulation. The reagent vial holder has nine positions for reagent vials. There are six positions for 100 µL to 4 mL vials and three for 1–16 mL. Gas-phase reactions are conducted on R1 and R3 with the support of suitable resins. Liquid phase reactions are conducted on R2. R2 can be heated to a controllable temperature to perform the synthesis. The reactors (R1, R3, and R4) are filled with different beads depending on the chemistry and purification. The formulation chamber is connected to R4 and has a final outlet for the final product collection.

#### 4.2.1. Beads and Bead’s Filling

Suitable beads are filled into the suitable reactor in the cassette according to the synthesis requirements and type of radionuclides used. For the radiosynthesis of ^11^C radiotracers, we are using C18 beads for both reaction and SPE purification. The beads are filled during cassette manufacturing by gravity, then pushed with compressed air, and helped to settle with ultrasonic vibration before final testing, packaging, and gamma irradiation.

#### 4.2.2. The Flow Resistance Measurement on R1 (Pressure Drop Test)

The flow resistance was measured in terms of the pressure drop on R1 with a given pressure value using an empty vial on EV A or B through R1 to the waste. We tested the flow resistance on R1, and these tests indicated the arrangement of the beads in the reactor chamber. If the beads in R1 were not packed compactly, this resulted in a lowered flow resistance, and the liquid was easily pushed through reactor R1. On the contrary, having beads arranged compactly, led to a higher resistance to flow than the optimized condition.

### 4.3. Radiosynthesis of [^11^C]Flumazenil and [^11^C]L-Deprenyl

[^11^C]CH_4_ (~30 GBq) was produced from the methane target via the ^14^N(p, α)^11^C nuclear reaction in a cyclotron. The target was filled with nitrogen gas mixed with 10% hydrogen and bombarded for 27–30 min at 35 µA. [^11^C]CH_3_I synthesis was performed the same as the previously reported method [36,37] using in-target produced [^11^C]CH_4_. For [^11^C]CH_3_OTf production, [^11^C]CH_3_I was passed through a silver triflate column (silver triflate impregnated with carbon powder) at 180 ^o^C. After the production of [^11^C]CH_3_I (~15 GBq) or [^11^C]CH_3_OTf (~13 GBq) was transferred through a separate line directly connected to the iMiDEV™ radiosynthesizer.

Different precursors were loaded on reactor R1 filled with C18 beads for [^11^C]flumazenil and [^11^C]*L*-deprenyl synthesis (Figure 4). [^11^C]CH_3_I or [^11^C]CH_3_OTf was passed through R1 at room temperature. After [^11^C]CH_3_I/[^11^C]CH_3_OTf was trapped on R1 (flow rate was 8 mL for flumazenil and 15 mL for deprenyl), an appropriate reaction time was followed. The reaction mixture was eluted from R1 and injected into a semi-preparative HPLC column for purification. The desired product from the semi-preparative HPLC was collected in the dedicated product loop. It was then injected into the microfluidic cassette for formulation using solid-phase extraction (SPE) cartridge R4. One of the vials in the cassette was used for product dilution in water to decrease the organic solvent concentration. Later, the product was pushed from the vial via R4 to the waste to trap the product. After trapping the product, it was washed with water. Finally, the product was eluted and diluted in the formulation chamber, and the final product was collected in a sterile vial through a sterile filter.

Only reactor R1 was explored for the radiochemical reaction. Required reagents were placed on the vial holder in the following vial positions. For HPLC injection, vial B was filled with 800 µL of 65% ethanol; vial E was filled with 100 µL (200 µg) of precursor in a 1:1 mixture of DMSO and DMF with powdered KOH for flumazenil and a 2:1 mixture of methanol and acetonitrile with PMP base for deprenyl. Vial F was filled with 400 µL of 100% ethanol for final product elution from R4. Vial G was filled with 5 mL of 0.9% saline or phosphate-buffered saline (PBS) for formulation. Vial H was filled with 10–12 mL of sterile water for product dilution from the HPLC, and vial I was filled with 2 mL of water to wash reactor R4.

After the synthesis starts, three main steps are carried out in the overall synthesis. The three main steps are, (A) radioactivity transfer, (B) HPLC purification, and (C) formulation.

Radioactivity transfer ([^11^C]CH_3_I or [^11^C]CH_3_OTf): After pressurizing reagent vials, the precursor was loaded on R1 from vial E by opening microfluidic valves V20, V18, V13, V12, V7, and V5 (Figure 5). The figure shows that microfluidic channels are used for precursor loading, and similarly, other microfluidic paths are used for reagent transfer. After the precursor is loaded, the system waits for radioactivity transfer by opening valves V20, V18, V13, V12, and V8. When radioactivity reached the cassette, it was detected by the sensor above R2. Then, valve V8 was manually turned off, and V3 and V7 were turned on to direct radioactive gas towards R1. It was allowed to pass [^11^C]CH_3_I or [^11^C]CH_3_OTf on R1 via the same microfluidic path as the precursor was loaded. When the maximum activity was trapped on R1, the reaction time was 3 min for flumazenil, and the deprenyl reaction was spontaneous.
Figure 5Precursor loading on the R1 from the vial E and other reagents placed on the respective positions.
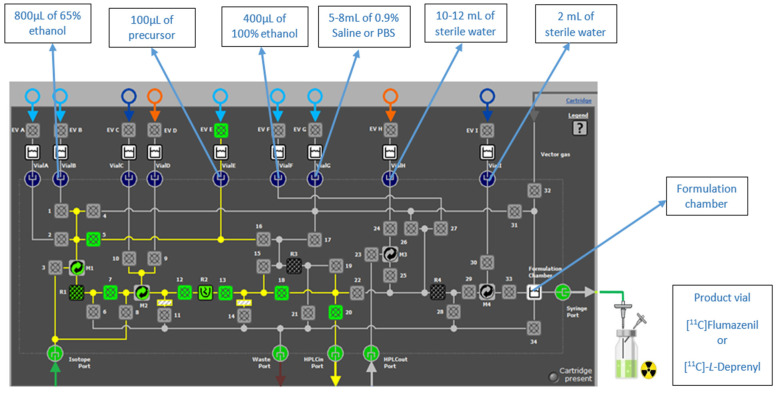
HPLC purification: After the reaction, trapped radioactivity was injected into a 1 mL HPLC injection loop (RV1) by opening microfluidic valves V20, V18, V13, V12, V7, and V1 from vial B (Appendix A). When fluid movement was detected from the optical sensor, RV1 was triggered to switch to position A (load) from position B.

When all the liquid had gone through the sensor, RV1 was again automatically triggered to switch back to position B (inject). After HPLC purification, the product was manually collected in the collection loop RV2 (2–5 mL) by switching position from A to B (Appendix A) and switched back to the original position after collection. It can also be done automatically by setting the detection value of the product’s peak height.

The semi-preparative HPLC system consisted of a reverse-phase (RP) ACE column (C18, 10 × 250 mm, 5 µm particle size), Knauer Azura pump P 6.1 L, Knauer Azura UV detector (UVD 2.1 S) and Capteur radio detector with CsI(TI) scintillation crystal used for radioactivity detection. The [^11^C]*L*-deprenyl was eluted with a mobile phase of 50% acetonitrile in phosphate buffer pH−9 with a flow rate of 8 mL/min, which gave a radioactive fraction corresponding to pure tracer with a retention time (t_R_) of 7–8 min. For [^11^C]flumazenil, the product was eluted with 35% of acetonitrile and 65% of 10 mM phosphoric acid (H_3_PO_4_) with a flow rate of 4 mL/min and retention time of approximately 7–8 min.

C.Formulation: After the product was collected, it was pushed back to vial H from the RV2 loop (position A) by opening valves V24, V23, and EV J (Appendix A). During this time, the same pressure was maintained on both sides of vial H and EV J. After the dilution of organic solvent in the collected product in vial H, it was trapped on R4 (C18 beads) by opening valves V28, V25, and V24 for further purification and formulation. Then, R4 was washed with 2 mL of sterile water from vial I by opening valves V20, V22, V29, and V30. Later, R4 was washed with 100% ethanol from vial F to elute the product into the formulation chamber by opening valves V34, V33, V29, and V27, followed by saline or PBS from vial G, by opening valves V34, V33, V29, and V26. The product was pushed to the sterile 10 mL product vial via a sterile filter from the formulation chamber.

Before the formulation tests, radioactivity trapping on R1 and radiochemical conversion were optimized using various solvents and bases for [^11^C]flumazenil and [^11^C]*L*-deprenyl. After the optimization of the reaction conditions, the entire synthesis was performed. All synthesis steps (radiochemical reaction, HPLC purification, and SPE purification and formulation) performed on the microfluidic cassette are shown in the flow chart (Appendix A)

The displacement of radioactivity in the cassette was detected from the sensors above R1, R2, R3, and R4 and monitored throughout the synthesis. This is shown in Figure 6. The product activity was measured using a dose calibrator, and the vial was weighed to calculate the total volume of the product. Quality control (QC) tests were performed to assess the radiochemical purity and identity of the final product.

### 4.4. Quality Control

The appearance test was performed visually by checking the product vial for particles or turbulence in the final volume. The pH of the product was determined using Merck pH paper (pH 4–9). The radiochemical purity, radiochemical identity, and radiochemical stability of [^11^C]flumazenil and [^11^C]*L*-deprenyl were analyzed by Agilent analytical high-performance liquid chromatography (HPLC). This HPLC is equipped with a quaternary pump G1311A, manual injector, ultraviolet (UV) detector G1314D, and radio detector. An analytical HPLC column from Xbridge (C18, 4.6 × 150 mm, 5 µm particle size) was used, and an isocratic mobile phase of 25% of acetonitrile and 75% of 10 mM phosphoric acid was used for [^11^C]flumazenil. A mobile phase of 50% acetonitrile and 50% ammonium phosphate buffer pH–9 was used for [^11^C]*L*-deprenyl. The flow rate was 2 mL/min. The identification of products was determined by co-injecting with the corresponding reference standard.

## 5. Conclusions

Radiosynthesis of [^11^C]flumazenil and [^11^C]*L*-deprenyl was explored with a newly developed microfluidic cassette based on a fully automated microfluidic batch-type radiosynthesizer. The produced radioligands comply with cGMP guidance, and these radiotracers can be readily used in clinical and preclinical applications in a single dose or DOD model. Our study addressed some of the limitations of conventional radiosynthesizers and existing microfluidic radiosynthesis techniques for radioligand production. We utilized less precursor for the reaction, integrated a semi-preparative HPLC system and formulated the final product in the iMiDEV module. For the first time, we explored a cassette-based batch-type microfluidic module for radiopharmaceutical drug production. In the future, our work will focus on further reduction of precursor quantity and synthesis time with an upgraded design of the microfluidic cassette.

## Figures and Tables

**Figure 1 molecules-27-08843-f001:**
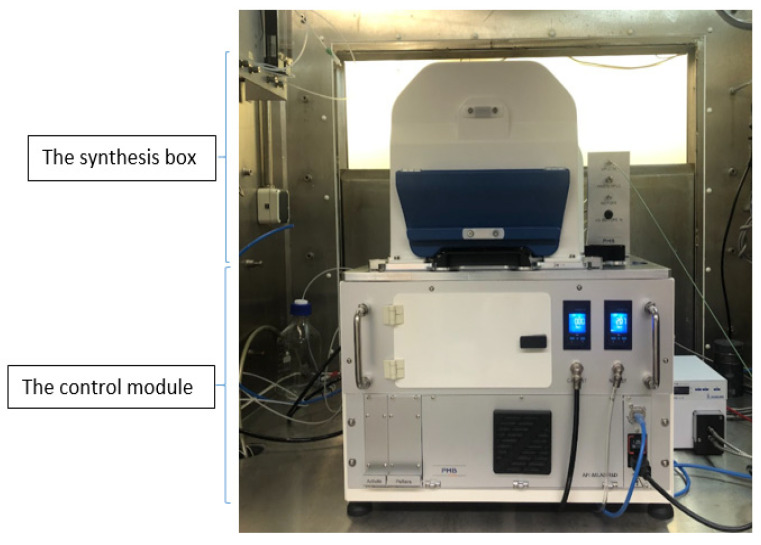
iMiDEV™ automated microfluidic radiosynthesizer.

**Figure 2 molecules-27-08843-f002:**
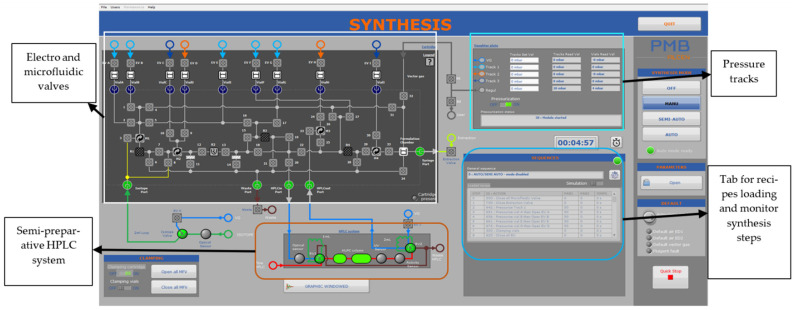
The detailed view of iMiDEV™ supervision software view (HMI).

**Figure 3 molecules-27-08843-f003:**
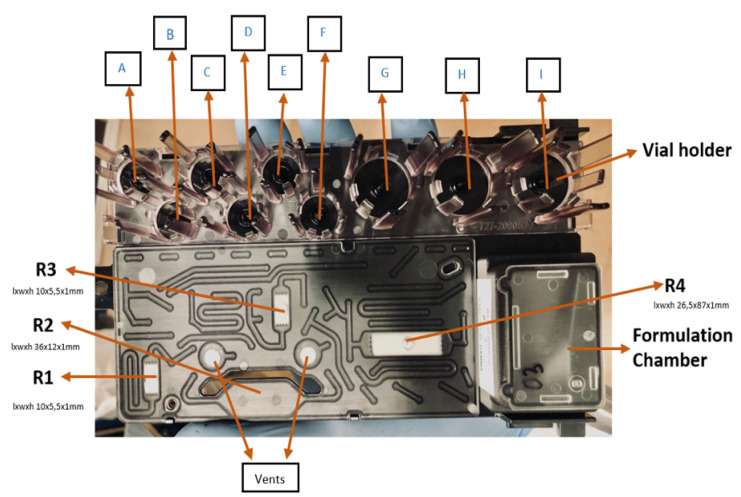
iMiDEV™ microfluidic cassette: Reactors (R1–R3 for reaction and R4 for SPE purification) and vial’s positions (A to F for 4 mL vials and G to I for 15 mL vials).

**Figure 4 molecules-27-08843-f004:**
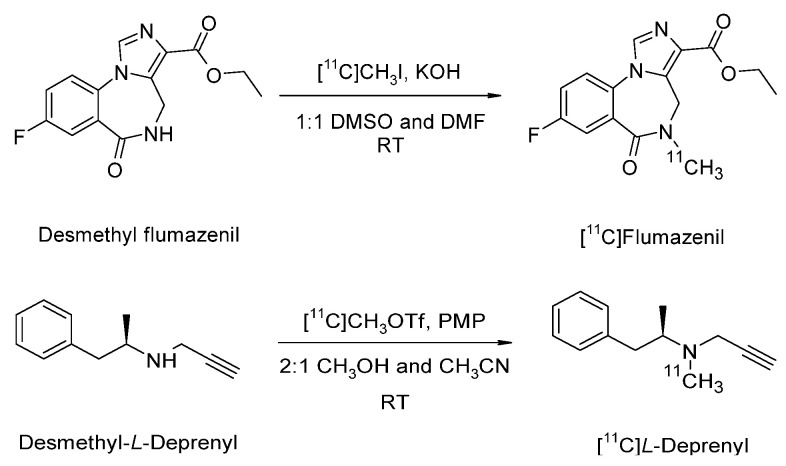
Reaction schematic for [^11^C]flumazenil and [^11^C]*L*-deprenyl synthesis.

**Figure 6 molecules-27-08843-f006:**
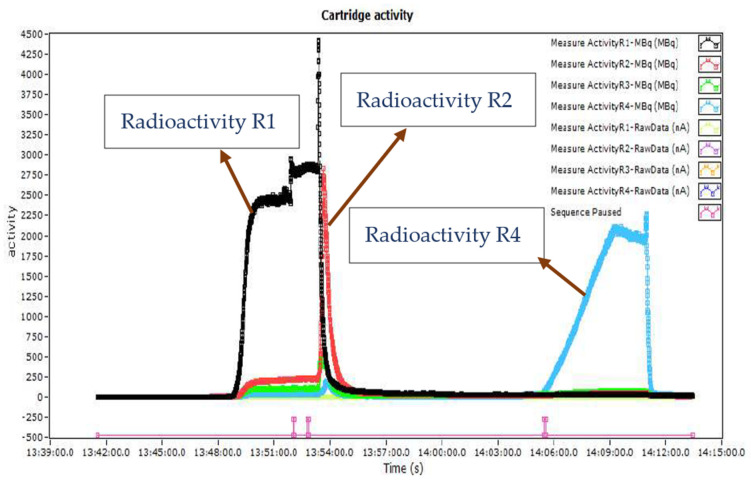
Recorded radioactivity data from R1, R2, R3 and R4 reactors from radio detectors during the complete radiosynthesis.

**Table 1 molecules-27-08843-t001:** Summary of [^11^C]flumazenil and [^11^C]*L*-deprenyl production.

Radioligands	Number of Production (*n*)	Product Activity (MBq)	Radiochemical Purity (%)	Molar Activity (GBq/µmol)
[^11^C]flumazenil	7	1644 ± 504	>99.0%	1912 ± 552
[^11^C]*L*-deprenyl	3	533 ± 20	>97.7%	1463 ± 439

**Table 2 molecules-27-08843-t002:** Summary of quality control tests for [^11^C]flumazenil and [^11^C]*L*-deprenyl.

Test	Acceptance Criteria	[^11^C]Flumazenil	[^11^C]*L*-Deprenyl
Appearance	Clear, colorless, free of particles	Complies	Complies
pH	5.0–8.0	5.5	5.5
Filter integrity test (bar)	>3.5 bar	3.8 bar	3.7 bar
Radiochemical purity	>95% by HPLC	>99%	>97.7%
Radionuclidic identity	T_1/2_ = 19.9–20.9 min	20.12 ± 0.11	20.12 ± 0.11
Radiochemical identity	Rt radio–Rt UV reference = (0.09–0.11) min	0.094	0.100
Residual solvents	Acetonitrile < 410 ppm Methanol < 3000 ppm Ethanol < 10% * DMSO < 5000 ppm and * DMF < 880 ppm	* Complies	Complies
Bacterial Endotoxin test	˂175 EU per dose †	<4 EU	<4 EU
Sterility	Sterile	Sterile	Sterile

Rt—retention time; *—Applies only for [^11^C]flumazenil; EU—Endotoxin units; †—The total volume(~8 mL).

## Data Availability

The data presented in this study are available on request to the corresponding authors.

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
