# Peer review of "Production of [11C]Carbon Labelled Flumazenil and L-Deprenyl Using the iMiDEV™ Automated Microfluidic Radiosynthesizer"

_molecules, 2022, doi:10.3390/molecules27248843_

Round 1

Reviewer 1 Report

This study assesses the use of a microfluidic cassette within an automated microfluidic radiosynthesizer, the iM-iDEV, to produce 11C-labelled drugs (flumazenil and L-deprenyl). The novelty of this particular study is the adaptation of the microfluidic cassette and radiosynthesizer system to enable radiolabeling with gaseous radionuclides such as C-11. The authors appear to have performed rigorous testing of the radiosynthesis unit and the microfluidic cassette to confirm its ability to adapt to these radiolabeling reaction requirements.

The reactors are filled with different beads depending on what process needs to be performed and the method development needed. Some additional information on how the beads are filled into the reactor chambers would be useful since this seems to be a point of error in the discussion section.

The authors attempt to give a good discussion of the pros and cons of the microfluidic adaption for this application discussing the various problems they had overcome as they reached a successful utilization of the device that yielded the results presented in the results section. The discussion, however, seems somewhat disjointed from the results section and is written a less formal manner. The information is good, but the authors should consider a better way to present this discussion and tie it to the results and conclusion. It also sounds like the study is ongoing and not complete for the publication e.g., lines 343-345.

In the conclusion, it is not clear why from the study presented in this manuscript, the authors are referencing the desire to test liquid phase reaction in the R2 reactor chamber under elevated temperature conditions. This does not fit well with the conclusions.

Line 143/144: it would help to reference where in the figure these solenoid valves are located.

It would be useful to have a discussion of cost and usefulness for the scientific community.  How much do the cassettes cost? And how easy are they use/reuse? How easy would it be for a standard lab to purchase one of these radiosynthesizers?

Author Response

Dear Reviewer 1,

Please find the attached file. I have added the response to your comments on the last pages (16 to 18) of the revised manuscript.

Thanks and best regards,

Hemantha Mallapura

Reviewer 2 Report

the manuscript titled "Production of [11C]carbon labelled flumazenil and L-deprenyl using the iMiDEV™ automated microfluidic radiosynthesizer" provides very optimized process of radio labeling.

however, the study does not provide a clear advantage over previous methods.

also, authors must provide qc hiply data as well as more experiments to add novelty  to the work.

Author Response

Dear Reviewer,

Please find the attached file. I have added response to your comments on the last pages (16 to 18) of the revised manuscript.

Thanks and best regards,

Hemantha Mallapura

Round 2

Reviewer 2 Report

The synthesis looks great with very high molar activity and radiochemical purity.

1) can authors add a schematic reaction of the radiolabeling process.

2) the pH in the range of 4 (in qc section) would be little less than the pharmacological acceptance limit of 5. 3) Did authors make any observation of low yield with  [ 11C]L-deprenyl synthesis in traditional methods. 4) is the low yield of the  [11C]L-deprenyl can be attributed to impure 11C-MeOTf. 5) could authors provide a semipreparative HPLC chromatogram for both synthesis.

Author Response

Dear reviewer,

Thanks, and kind regards,

Hemantha Mallapura
